# HyperINR: Ensuring Semantics in Weights with Implicit Function Theorem

## Abstract

Implicit Neural Representations (INRs) have demonstrated remarkable capability in representing 2D and 3D data, whose semantics is convincingly captured in the weights of the corresponding neural network. Despite successes in applying INRs to various applications, a precise theoretical explanation for the mechanism of encoding semantics of data into network weights is still missing. In this work, we propose a jointly trainable *HyperINR* model, which learns a hypernetwork to map a learnable low-dimensional latent space to the weight space of an INR. By employing the Implicit Function Theorems, the *HyperINR* is shown to be a mathematically rigorous framework that ensures the mapping of the semantics of data to the weight latent representations. Extensive experiments of classification tasks on 2D and 3D data confirm the effectiveness of our approach and demonstrate superior performance compared to state-of-the-art methods.

## 1 Introduction

With the rapid development of Artificial Intelligence (AI), we are witnessing an explosion in the number and scale of neural networks. Despite their remarkable generalization capabilities, many of these models are often trained on overlapping benchmarks or designed to perform similar functionalities. Models trained on similar data would arguably share similar underlying relationships at the level of their learned weights. Recent work, such as the "model atlas", leverages model weights from platform Hugging Face to visualize relationships among individual models and entire populations (Horwitz et al., 2025). Building on this perspective, network weights can be viewed as a structured data modality. Namely, after the neural networks are trained, their weights can encode semantics of the training data. In this work, we investigate how the semantics of data can be embedded in network weights, and how to guarantee that the captured representations faithfully reflect the original training data.

The concept of neural network weights being capable of encoding data semantics can be traced back to Bayesian neural networks, where weights capture a posterior distribution that reflects the model's learned semantics of data (Gal & Ghahramani, 2016). Such a perspective benefits tasks like meta-learning (Finn et al., 2017), style transfer (Gatys et al., 2016; Karras et al., 2019), and efficient fine-tuning (Houlsby et al., 2019). Methods, like HyperDreamBooth (Ruiz et al., 2024) and HyperTuning (Phang et al., 2023), show that modifying only a small subset of weights generated by hypernetworks (Ha et al., 2017) can substantially alter models' behavior. Other techniques, such as model merging (Yang et al., 2024), aim to integrate knowledge across multiple models by fusing their weights for better performance. More recently, ICLR 2025 Workshop on Weight Space Learning (WSL) has further consolidated this direction (Schürholt et al., 2024) and highlights directions ranging from treating weights as a data modality to developing new methods for analyzing and synthesizing models in weight space.

Among various Neural Network models, Implicit Neural Representations (INRs) (Sitzmann et al., 2020) provide a convenient medium for such investigations. By parameterizing data samples as coordinate-based functions (Park et al., 2019), INRs leverage neural networks' universal approximation ability (Kidger & Lyons, 2020) to succeed in reconstructing a wide variety of signals. Each data sample is associated with its own neural network, whose weights serve as an encoding of the underlying semantics. This makes INRs a natural testbed for analyzing how individual samples are embedded into weight spaces (Navon et al., 2023; Zhou et al., 2023a).

In this work, we propose a training pipeline that constrains the INR model within a low-dimensional latent space. Latent embeddings are mapped through a dataset-shared hypernetwork to generate INR parameters, which are then used to reconstruct individual 2D images or 3D shapes. In this formulation, every input has its own weights derived from its latent embedding vector, while both the latent vectors and the hypernetwork are optimized jointly during training. Such a framework is grounded in the *Implicit Function Theorem* (IFT), which guarantees, under certain conditions, the existence of an explicit mapping between implicitly related variables. Our use of the IFT complements prior progress on INR classifier and semantics-in-weights providing the missing theoretical foundation for a more complete understanding of data-weight relationships.

In experiments across 2D data and 3D shape classification tasks, we observe that the jointly optimized embeddings form distinct clusters in both latent and weight spaces. Our outperforming classification accuracy further verifies numerically that the latent weight embeddings faithfully capture these semantics. Compared with prior work, our approach employs minimal data pre-processing and parsimonious pipeline, which highlights the potential effectiveness of this proof-of-concept paradigm.

In summary, our contributions are threefold:

1. The work introduces a jointly trainable HyperINR model, which learns a hypernetwork to map a learnable low-dimensional latent space to the weight space of an INR. This architecture yields a latent weight representation that compactly encodes data semantics and provides the theoretical guarantee for using "weights as a modality" in downstream tasks such as classification.

2. Rigorous analysis is conducted to connect semantics of data, network weights and latent weight representations through the IFT.

3. Experiments across 2D and 3D datasets show consistent outperformance over existing INR-based classifiers.

## 2 RELATED WORK

**Equivariance and symmetry in INR weights**   Using INR weights directly for downstream tasks, such as classification, can be challenging, since the optimized INRs can vary significantly depending on their initializations, even for the same signal (Huang et al., 2025). Permutation-equivariant classifiers, such as DWS (Navon et al., 2023) and NFN (Zhou et al., 2023a), address these parameter symmetry issues by ensuring that models are invariant to weight reordering. Similarly, Transformer-based architectures (Zhou et al., 2023b) can also produce permutation-equivariant representations of weights for downstream tasks. More recently, MWT (Gielisse & van Gemert, 2025) introduces equivariant INR classifiers with end-to-end supervision, achieving strong performance but in a setting where labels are integrated into the representation. EquiGen (Huang et al., 2025) focuses on learning generative models over weight permutations of functionally similar networks, enabling INR synthesis from limited data. While these approaches make progress in handling symmetry, a more fundamental challenge remains unresolved. Namely, optimizing Neural Networks even for the same data samples can not be guaranteed to converge to the same set of weights.

**Latent representations for weights**   Another line of works embeds INR weights into low-dimensional latent spaces. Earlier INR works, such as SIREN (Sitzmann et al., 2020), have already explored hypernetworks, where weights are generated from embeddings conditioned on inputs. Building on this idea, D'OH (Gordon et al., 2024) proposed a decoder-only hypernetwork in which embeddings are directly optimized from random initializations, without assuming structure in the embedding space. From an architectural perspective, many works also adopt hypernetwork-based INR formulations to improve generalization and reconstruction (Gu et al., 2023; Gu & Yeung-Levy, 2025; Chen & Wang, 2022; Kim et al., 2023). However, they do not focus on data semantics preservation in the network weight space. Functa (Dupont et al., 2022) formalizes the view of INRs as data: a shared base network captures common semantics, while per-sample modulation vectors encode variations. These low-dimensional modulation vectors can also serve as useful inputs for downstream tasks. Schürholt et al. (2022) train Hyper-representations with self-supervised learning on the weights of INR model zoos to obtain embeddings for downstream tasks. Similarly,

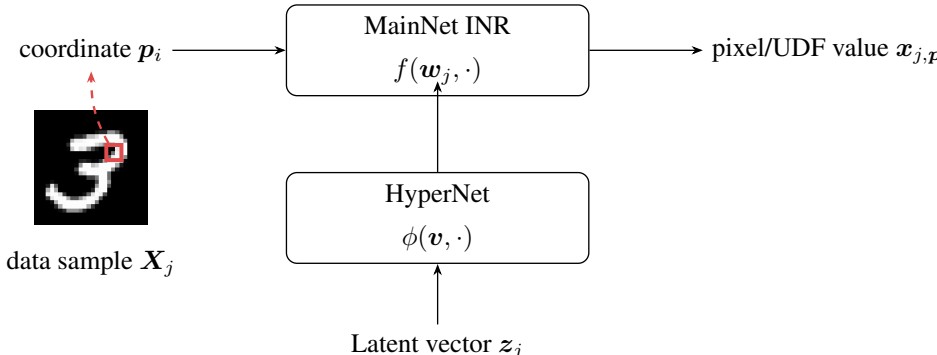

Figure 1: HyperINR: A hypernetwork generates weights $\boldsymbol{w}_j$ from learnable latent vector $\boldsymbol{z}_j$, which are then used by the main network $f(\boldsymbol{w}_j, \cdot)$ to map coordinates to pixel or UDF values.

inr2vec (Luigi et al., 2023) maps INR weights into latent embeddings for classification and generation, showing that semantic structure is preserved in such representations.

Our proposed framework is related to these latent embedding assumptions and hypernetwork-based architectures but differs in an important respect. Rather than conditioning embeddings on raw inputs or first fitting INR weights, and then learning a reduced representation, we directly optimize latent embeddings through a shared hypernetwork to reconstruct the data. This design is more direct and parsimonious, revealing the underlying weight structure in a natural way, while avoiding intermediate ambiguities.

## 3 METHODOLOGY

### 3.1 THE HYPERINR PIPELINE

Instead of learning low-dimensional representations from the weights of individually pretrained INRs for all data samples, we propose to utilize a hypernetwork which maps a learnable low-dimensional latent space to the weight space of INR main networks. The HyperINR pipeline is shown in Fig. 1.

Let $\mathcal{P} \subset \mathbb{R}^p$ be an open set of $p$-dimensional coordinates and $\mathcal{W} \subset \mathbb{R}^d$ be the set of weights of INRs. An INR main network takes sampled coordinates $\boldsymbol{p} \in \mathcal{P}$ as input and generates the corresponding pixel value or unsigned distance function (UDF) value (Mullen et al., 2010) in $\mathbb{R}^c$. Specifically, for a set of sampled coordinates $\{\boldsymbol{p}_i\}_{i=1}^n \subset \mathcal{P}$, any data example $\boldsymbol{X} := \{\boldsymbol{x}_i\}_{i=1}^n \in \mathcal{X} \subset \mathbb{R}^{n \times c}$ can be represented as a sampled mapping from coordinate $\boldsymbol{p}_i$ to data value $\boldsymbol{x}_i$. Here, we define an INR main network as the following mapping

$$f: \mathcal{W} \times \mathcal{P} \to \mathbb{R}^c, \qquad (\boldsymbol{w}, \boldsymbol{p}) \mapsto f(\boldsymbol{w}, \boldsymbol{p}). \tag{1}$$

In this work, we assume that the set of weights of INRs admits a low-dimensional structure. Specifically, the weights $\boldsymbol{w}_i$ for each data sample are generated by a hypernetwork, which maps a learnable latent embedding $\boldsymbol{z}_j \in \mathcal{Z} \subset \mathbb{R}^l$ to the weights $\boldsymbol{w}_i \in \mathcal{W}$ of the main network, i.e.,

$$\phi: \mathcal{V} \times \mathcal{Z} \to \mathcal{W}, \qquad (\boldsymbol{v}, \boldsymbol{z}_j) \mapsto \phi(\boldsymbol{v}, \boldsymbol{z}_j), \tag{2}$$

where $\mathcal{V} \subset \mathbb{R}^k$ denotes the space of hypernetwork weights, which are shared across all data examples. For a given set of data samples $\{\boldsymbol{X}_j\}_{j=1}^t$, a common training strategy is to find an optimal set of weights of the hyper-network $\boldsymbol{v}^* \in \mathcal{V}$ and a set of latent embeddings $\{\boldsymbol{z}_j^*\}_{j=1}^t \subset \mathcal{Z}$, such that the corresponding INRs exactly represent the data samples, i.e., for all $i = 1, \ldots, n$ and $j = 1, \ldots t$, the following equation holds true

$$f(\phi(\boldsymbol{v}^*, \boldsymbol{z}_j^*), \boldsymbol{p}_i) = \boldsymbol{x}_{j, \boldsymbol{p}_i}, \tag{3}$$

with $\boldsymbol{X}_j := \{\boldsymbol{x}_{j,i}\}_{i=1}^n \in \mathcal{X}$. With such a construction, it is clear that the latent embeddings $\boldsymbol{z}_j^*$ can be considered as a point-wise representation of the original data samples $\boldsymbol{X}_j$. The core question investigated in this work is how to ensure that semantics of data are captured in the weights of corresponding INRs.

## 3.2 ENSURING SEMANTICS IN WEIGHTS WITH THE IMPLICIT FUNCTION THEOREM

In this subsection, we first investigate the residual loss for training the HyperINR, and then apply the IFT to analyze the global minimum of the loss function to ensure the capture of semantics of data into weights of INRs.

In our analysis, we assume that the data set $\mathcal{X} \subset \mathbb{R}^{n \times c}$ is a compact local-dimensional manifold, and the space of latent embeddings $\mathcal{Z} \subset \mathbb{R}^l$ is an open set. The classic residual loss for assessing the reconstruction of the INR against a given data sample can be defined as

$$\ell \colon \mathcal{V} \times \mathcal{Z} \times \mathcal{X} \to \mathbb{R}, \quad \ell(\boldsymbol{v}, \boldsymbol{z}, \boldsymbol{X}) := \tfrac{1}{2} \left\| \{ f(\phi(\boldsymbol{v}, \boldsymbol{z}), \boldsymbol{p}_i) \}_{i=1}^n - \boldsymbol{X} \right\|_F^2$$
$$= \tfrac{1}{2} \sum_{i=1}^n \left\| f(\phi(\boldsymbol{v}, \boldsymbol{z}), \boldsymbol{p}_i) - \boldsymbol{x}_{\boldsymbol{p}_i} \right\|_2^2, \tag{4}$$

where $\boldsymbol{x}_{\boldsymbol{p}_i}$ denotes the value of sample $\boldsymbol{X}$ at coordinate $\boldsymbol{p}_i$, and $\| \cdot \|_F$ denotes the Frobenius norm of matrices. Let us assume that the size of the INRs is sufficiently large to ensure exact reconstruction in the whole data manifold $\mathcal{X}$, i.e., there is a hypernetwork weight $\boldsymbol{v}^* \in \mathcal{V}$, which ensures exact reconstruction of any sample $\boldsymbol{X} \in \mathcal{X}$. Consequently, for any pair $(\boldsymbol{z}, \boldsymbol{X}) \in \mathcal{Z} \times \mathcal{X}$, the gradient of $\ell$ vanishes at $(\boldsymbol{v}^*, \boldsymbol{z}, \boldsymbol{X})$.

With a fixed $\boldsymbol{v}^*$ for the hypernetwork, we define the following function

$$\ell_{\boldsymbol{v}^*} \colon \mathcal{Z} \times \mathcal{X} \to \mathbb{R}, \qquad (\boldsymbol{z}, \boldsymbol{X}) \mapsto \ell(\boldsymbol{v}^*, \boldsymbol{z}, \boldsymbol{X}). \tag{5}$$

Let us denote $\boldsymbol{w} := \phi(\boldsymbol{v}^*, \boldsymbol{z})$, and $\boldsymbol{\epsilon}_i := f(\phi(\boldsymbol{v}^*, \boldsymbol{z}), \boldsymbol{p}_i) - \boldsymbol{x}_{\boldsymbol{p}_i}$. We compute the differential map of $\ell_{\boldsymbol{v}^*}$ with respect to the hyperparameter $\boldsymbol{z}$ as

$$\mathrm{D}_1 \, \ell_{\boldsymbol{v}^*}(\boldsymbol{z}, \boldsymbol{X}) = \sum_{i=1}^n \boldsymbol{\epsilon}_i^\top \, \mathrm{D}_1 \, f(\boldsymbol{w}, \boldsymbol{p}_i) \, \mathrm{D}_2 \, \phi(\boldsymbol{v}^*, \boldsymbol{z}), \tag{6}$$

where $\mathrm{D}_1 \, f(\boldsymbol{w}, \boldsymbol{p}_i)$ is the differential map of $f$ with respect to the first variable, and $\mathrm{D}_2 \, \phi(\boldsymbol{v}^*, \boldsymbol{z})$ is the differential map of $\phi$ with respect to the second variable. For any pair $(\boldsymbol{z}, \boldsymbol{X}) \in \mathcal{Z} \times \mathcal{X}$, we have the vanishing gradient as

$$\mathrm{D}_1 \, \ell_{\boldsymbol{v}^*}(\boldsymbol{z}, \boldsymbol{X}) = 0. \tag{7}$$

Clearly, it establishes a potential mapping between $\boldsymbol{z}$ and $\boldsymbol{X}$ in terms of implicit functions.

Let us define the following function

$$\xi_{\boldsymbol{v}^*} \colon \mathcal{Z} \times \mathcal{X} \to \mathbb{R}^l, \qquad (\boldsymbol{z}, \boldsymbol{X}) \mapsto (\mathrm{D}_1 \, \ell_{\boldsymbol{v}^*}(\boldsymbol{z}, \boldsymbol{X}))^\top, \tag{8}$$

which is essentially the gradient of $\ell_{\boldsymbol{v}^*}$ with respect to $\boldsymbol{z}$, by endowing the canonical Euclidean metric. We then investigate the properties of the Jacobian of $\xi_{\boldsymbol{v}^*}$ with respect to the first variable at $\boldsymbol{z}$, which is nothing but the second differential of $\ell_{\boldsymbol{v}^*}$ with respect to $\boldsymbol{z}$. Straightforwardly, we have

$$\mathrm{D}_1 \, \xi_{\boldsymbol{v}^*}(\boldsymbol{z}, \boldsymbol{X}) = \sum_{i=1}^n (\mathrm{D}_1 \, f(\boldsymbol{w}, \boldsymbol{p}_i) \, \mathrm{D}_2 \, \phi(\boldsymbol{v}^*, \boldsymbol{z}))^\top \, \mathrm{D}_1 \, f(\boldsymbol{w}, \boldsymbol{p}_i) \, \mathrm{D}_2 \, \phi(\boldsymbol{v}^*, \boldsymbol{z}). \tag{9}$$

Note, that the above expression is essentially the Hessian matrix of the residual loss $\ell$ with respect to the exact latent representation $\boldsymbol{z}$ of each sample $\boldsymbol{X}$ at a globally optimal hypernetwork weight $\boldsymbol{v}^*$. It is known that if $\mathrm{D}_1 \, \xi_{\boldsymbol{v}^*}(\boldsymbol{z}, \boldsymbol{X})$ is of full rank on $\mathcal{Z} \times \mathcal{X}$, then the value of zero is a regular value of the map $\xi_{\boldsymbol{v}^*}$. Furthermore, the global implicit function theorem (Krantz & Parks, 2002) implies that there is a unique smooth map $g \colon \mathcal{X} \to \mathcal{Z}$, which satisfies for all $\boldsymbol{X} \in \mathcal{X}$

$$\xi_{\boldsymbol{v}^*}(g(\boldsymbol{X}), \boldsymbol{X}) = 0. \tag{10}$$

By knowing $\mathrm{D}_1 \, f(\boldsymbol{w}, \boldsymbol{p}_i) \in \mathbb{R}^{c \times d}$, $\mathrm{D}_2 \, \phi(\boldsymbol{v}^*, \boldsymbol{z}) \in \mathbb{R}^{d \times l}$, and $c < l$, we see that the Jacobian of $\xi_{\boldsymbol{v}^*}$ is a sum of $n$ positive semi-definite symmetric matrices with their ranks being not larger than $c$. Therefore, a necessary condition to ensure the Jacobian of $\xi_{\boldsymbol{v}^*}$ to have full rank of $l$ is $nc \geq l$. This simple condition can be used as a guidance to determine the minimal number of coordinates that is needed to ensure semantics of data are captured in the weights of the corresponding INRs. Note,

that this is typically satisfied for both 3D shape and image data, where the dimension of latent vector $l$ is always much smaller than $nc$. Moreover, it is also worth noticing that the rank of the Jacobian of $\xi_{\boldsymbol{v}^*}$ depends further on various factors, such as geometry of the data manifold $\mathcal{X}$, the architectures of both main network and hypernetwork, and the choice of activation functions. Thus, investigation of sufficient conditions for the Jacobian being of full rank is considered to be a challenging research question in the future.

As shown in our analysis above, by leveraging the gradient of the residual loss $\ell$ as implicit constraints between the data manifold $\mathcal{X}$ and the latent space of weights of INRs $\mathcal{Z}$, global semantics of the data manifold can be captured in the latent space of INR weights, under the conditions of exact reconstruction of INRs. Practically, for a given finite number of samples $\{\boldsymbol{X}_j\}_{j=1}^t \subset \mathcal{X}$, we define the total residual loss function of all data samples as

$$\mathcal{L}(\boldsymbol{v}, \{\boldsymbol{z}_j\}_{j=1}^t) := \sum_{j=1}^t \ell(\boldsymbol{v}, \boldsymbol{z}_j, \boldsymbol{X}_j). \tag{11}$$

With minimizing the total residual loss, we assume a hypernetwork weight $\boldsymbol{v}^*$ that reconstruct exactly all the samples. If the Jacobian as in equation 9 at all samples has full rank, the classic IFT implies that there exist open neighborhoods $\mathcal{N}_{\boldsymbol{X}_j}$ containing $\boldsymbol{X}_j$ and $\mathcal{N}_{\boldsymbol{z}_j^*}$ containing $\boldsymbol{z}_j^*$, such that there is a unique continuously differentiable function $g \colon \mathcal{N}_{\boldsymbol{X}_j} \to \mathcal{N}_{\boldsymbol{z}_j^*}$, which satisfies

$$g(\boldsymbol{X}_j) = \boldsymbol{z}_j^*, \qquad \text{and} \qquad \mathrm{D}_1\, \xi_{\boldsymbol{v}^*}(g(\boldsymbol{X}), \boldsymbol{X}) = 0, \text{ for all } \boldsymbol{X} \in \mathcal{N}_{\boldsymbol{X}_j}. \tag{12}$$

Clearly, only local semantics around the samples is guaranteed to be captured in the neighborhood around their corresponding latent embeddings. Nevertheless, we hypothesize that a sufficiently large number of samples, which uphold the conditions for the local IFT, enhance the chance of global semantics of data being preserved in the latent space of INR weights. Our experimental results in the next section support this hypothesis.

## 4 EXPERIMENTS

### 4.1 TRAINING AND DEPLOYMENT OF HYPERINR FOR CLASSIFICATION

The common paradigm for INR classifiers and "weight as a modality" works typically involves two phases: first, training INRs to reconstruct data samples to transform semantics in the raw input data to INR weights or latent weight space; second, using the optimized weights for downstream tasks. We also follow this two-phase training pipeline shown in Fig. 2. The Phase A transformation process, which prepares the weight data, is performed in two steps: one for the training set and one for the testing set. In the first step, we prepare the training set by jointly optimizing the hypernetwork $\phi(\boldsymbol{v}, \cdot)$ and the latent embeddings $\{\boldsymbol{z}_i\}$ to minimize the reconstruction loss for all training samples. The latent embeddings are initialized randomly around the origin and updated during the optimization. The second step involves fixing weights $\boldsymbol{v}^*$ of the hypernetwork and inferring new embeddings for unseen test data by minimizing the reconstruction loss. Prior approaches require a separate stage for fitting the INRs and then transforming it to a low-dimensional space (Luigi et al., 2023; Zhou et al., 2023a). Our joint optimization of INR reconstruction and embedding is inherently more parallel-efficient.

To evaluate the effectiveness of our HyperINR framework and the learned latent embedding of weights, we conduct classifications on both 2D images and 3D shapes, following prior works (Navon et al., 2023; Zhou et al., 2023a;b; Luigi et al., 2023) in Phase B. We first investigate the natural clustering structure of the latent space, then assess classification accuracy using a simple MLP classifier.

We keep the network size tightly constrained in all experiments, aiming to validate the proposed framework under minimal rather than expansive model capacity. The hypernetwork is a two-layer Multi-layer perceptron (MLP) (Haykin, 1994) with multiple heads, where each head is a one-layer linear mapping. Each head generates the weights for a single INR layer, reshaped into the appropriate weight matrix and bias vector. The main network is a SIREN (Sitzmann et al., 2020), which is never trained directly. Its weights are always generated by the hypernetwork from latent embeddings. No special initialization or regularization is required, since the SIREN itself is not optimized directly. The dimension of the learnable latent vectors $\boldsymbol{z}$ is set to 20 for 2D datasets and

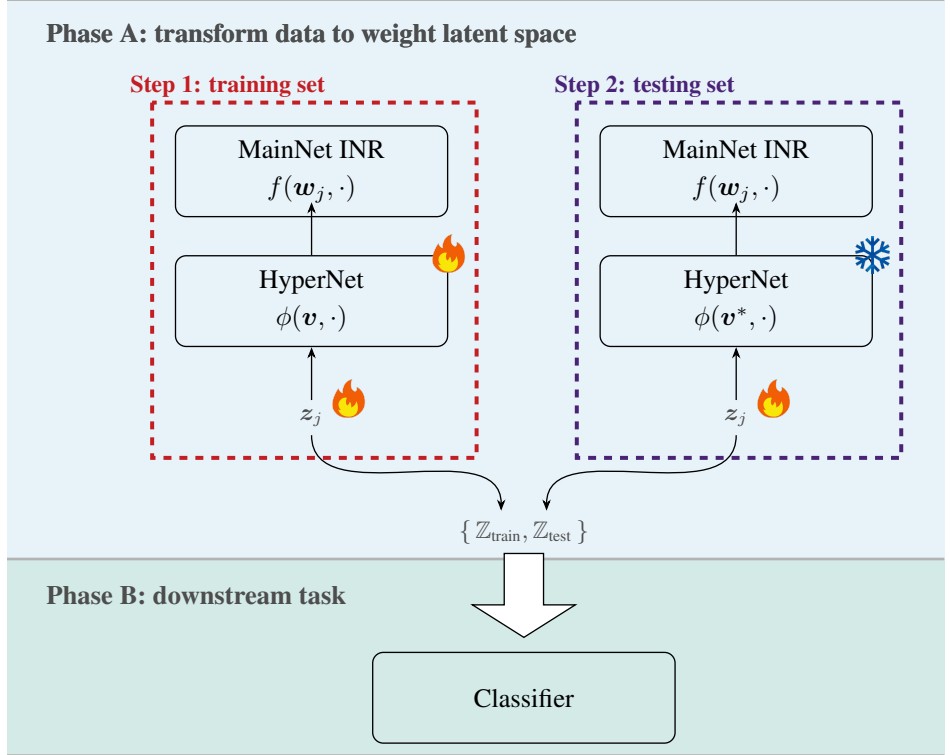

Figure 2: Phase A transforms data samples to latent weight representations using HyperINR, while Phase B utilizes these latent representations for downstream classification tasks.

10 for 3D datasets. In contrast, similar work like inr2vec (Luigi et al., 2023), which also makes a low-dimensional weight assumption, uses latent vectors with dimensions of 512 or 1024. More configurations are detailed in the Appendix.

We evaluate on five datasets: MNIST (Yann, 2010) and FashionMNIST (Xiao et al., 2017) for 2D images, and ModelNet40 (Wu et al., 2015), ShapeNet10 (Chang et al., 2015), and ScanNet10 (Qin et al., 2019) for 3D shapes. The 2D datasets can be directly used for INR reconstruction, whereas the 3D meshes require conversion to a coordinate-based representation (Mullen et al., 2010). Prior works often prepare up to 500K query points near surfaces and then sample different subsets of 10K points during training as data augmentation to improve INR robustness (Luigi et al., 2023). In contrast, we just uniformly sample 10K queries and obtain the ground truth UDF values via a KDTree (Ravi et al., 2020). This simple setup is adequate for validating our concept. More dataset details are provided in A.1.

The codes will be made publicly available upon paper acceptance.

### 4.2 CLUSTERING IN LATENT AND WEIGHT SPACE

We first examine whether the learned latent embeddings exhibit clustering aligned with semantic classes. Using ShapeNet10 as an example, we train on all 10 classes but visualize embeddings from three categories (chair, sofa, table) for clarity. For visualization, we apply Principal Component Analysis (PCA) (Pearson, 1901), a linear dimensionality reduction method, to avoid distortions introduced by nonlinear projections. As shown in Figure 3, the training embeddings form well-separated clusters, and test samples fall into the neighborhoods of their respective categories with the hypernetwork fixed. Such distinct clustering is already visible under our minimal setup. Using only the mean squared error (MSE) loss, a low-dimensional latent weight space assumption, and the guarantees provided by the IFT, the latent embeddings appear to preserve semantic information. Furthermore, PCA projections of the hypernetwork-generated weights exhibit similarly distinct clusters, reinforcing that semantics are preserved in both latent and weight spaces, as shown in Fig-

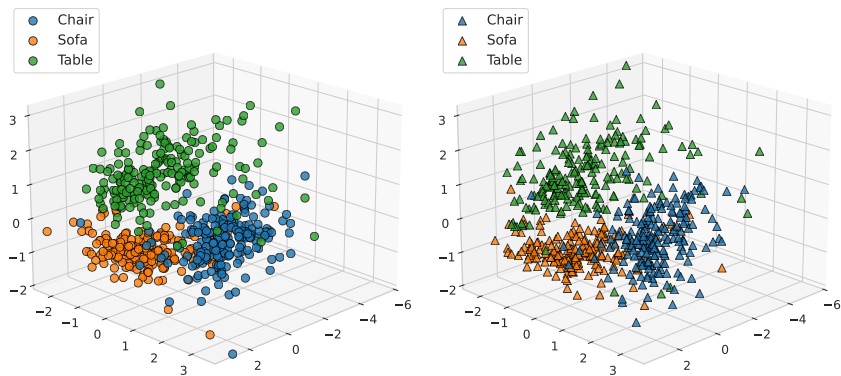

Figure 3: Distinct clustering of latent vectors $z$ from ShapeNet10 using 3D PCA. **Left:** Training samples form well-separated clusters. **Right:** Test samples fall into the neighborhoods of their respective categories under a fixed hypernetwork.

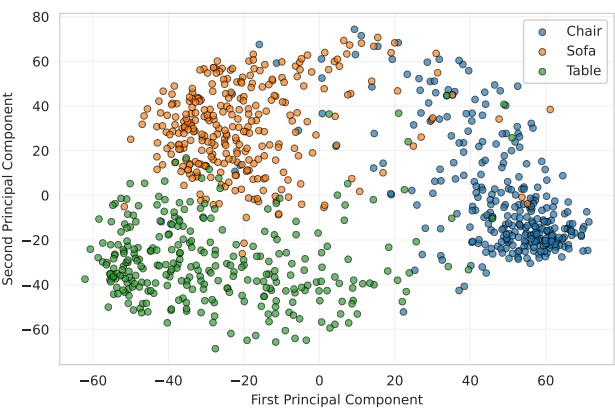

Figure 4: Distinct clustering of hypernetwork-generated weights for ShapeNet10 using 2D PCA.

ure 4. Achieving clean and unambiguous clusters in weight space remains difficult for previous work, suggesting that our lightweight setup offers a promising direction.

### 4.3 EVALUATION OF CLASSIFICATION ACCURACY UNDER A LIGHTWEIGHT SETUP

Based on the observation of distinct clustering in the latent space, we next quantitatively assess the quality of the learned representations by training a trivial MLP classifier on the latent embeddings. Classification accuracy is reported on five datasets, with our HyperINR achieving state-of-the-art results on three benchmarks and remaining competitive on MNIST and ScanNet10, as summarized in Tables 1 and 2.

Because our pipeline differs substantially from prior work in objectives, architectures, and data pre-processing, we do not re-run their implementations under our setting. Instead, we cite their published results. Although we do not establish a strict one-to-one comparison with prior methods, we aim for transparency and fairness by employing only a small MLP classifier and avoiding heavy data manipulations such as augmentation or additional engineering tricks. We report a single representative run without variance because our method is not affected by the typical initialization sensitivity of INR fitting. Since the latent vectors $z$ are initialized consistently near the origin and the SIREN weights are generated by a hypernetwork, our results aren't affected by weight initialization. The only variability comes from standard random-seed effects.

Table 1: Classification top-1 accuracy (%) on 2D datasets.

| Method | MNIST | FashionMNIST |
|---|---|---|
| inr2vec-arch (Navon et al., 2023) | $23.69 \pm 0.10$ | $22.33 \pm 0.41$ |
| DWS (Navon et al., 2023)) | $85.71 \pm 0.57$ | $67.06 \pm 0.29$ |
| NFN Zhou et al. (2023a) | $92.9 \pm 0.218$ | $75.6 \pm 1.07$ |
| Inr2Array (Zhou et al., 2023b) | $\mathbf{98.5 \pm 0.00}$ | $79.3 \pm 0.00$ |
| **Ours** | **98.18** | **87.76** |

Table 2: Classification top-1 accuracy (%) on 3D datasets

| Method | ModelNet40 | ShapeNet10 | ScanNet10 |
|---|---|---|---|
| NFN (Zhou et al., 2023a) | - | $88.7 \pm 0.461$ | $65.9 \pm 1.10$ |
| inr2vec (Luigi et al., 2023) | 87.0 | 93.3 | 72.1 |
| **Ours** | **87.2** | **93.5** | **71.0** |

On MNIST, Inr2Array reports higher accuracy, but uses data augmentation, a large transformer-based classifier, and heavy optimization ($\sim$22M parameters, $\sim$13 hours of training). In contrast, our setup employs a small MLP with about 1M parameters, mostly in the hypernetwork heads, which serve as the unavoidable linear mapping from low-dimensional vectors to the INR main network weights. Our joint learning converges in 30 minutes on a single NVIDIA A5000 (24GB RAM) and achieves comparable or better results. We provide more information regarding reproduction in A.3.

These classification experiments are intended as proof-of-concept demonstrations rather than comprehensive benchmarks. For example, we do not rebalance the class distribution in ShapeNet, since accuracy serves primarily as an indicator of the semantic fidelity of the latent embedding $z$. To maintain this focus, our method deliberately adopts a lightweight setup: a simple MLP classifier trained with the Adam optimizer and pure MSE loss, without AdamW, dropout, or L2 regularization, etc. Other works might rely on more complex designs. DWS (Navon et al., 2023) incorporates extensive data augmentation, while the NFN method (Zhou et al., 2023a) builds on 3 equivariant NF-layers with 128 channels, followed by 4 linear layers with 1,000 hidden neurons. Similarly, inr2vec (Luigi et al., 2023) employs a large encoder (four layers of 512, 512, 1024, and 1024 units) and a decoder with four hidden layers of 512 units each; their INR main network is also a sizable MLP with 4 hidden layers of 512 units, and they report that reducing its size compromises reconstruction quality.

Our design choices reflect an intention to isolate and validate the effectiveness of the proposed pipeline itself, rather than to depend on auxiliary techniques or large model capacity. The main contribution of this work is theoretical, and the experiments are presented to confirm the frameworks validity under restrained conditions.

### 4.4 SMOOTH INTERPOLATIONS IN LATENT SPACE

To further examine the quality of the learned latent space, we conduct linear interpolations between the latent embeddings of two ShapeNet samples. As shown in Figure 5, the interpolated latent embeddings $z$ yield reconstructions that not only achieve good fidelity to the original endpoints but also smoothly morph from one shape to the other. The absence of abrupt artifacts suggests that the latent space learned by HyperINR is continuous and semantically meaningful, rather than memorizing isolated training instances.

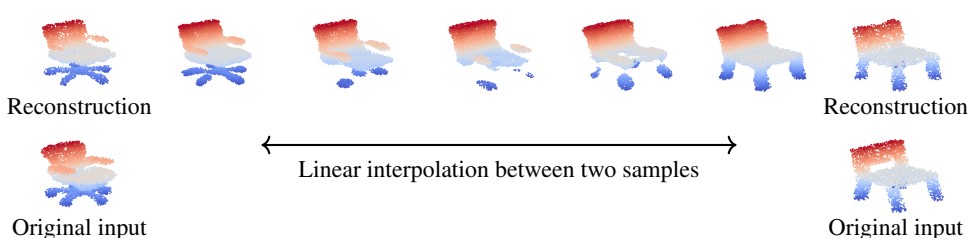

Reconstruction                                                                    Reconstruction

Linear interpolation between two samples

Original input                                                                    Original input

Figure 5: Linear interpolation in the latent space between two ShapeNet chair samples. The results exhibit smooth transitions in geometry, indicating that the latent space is continuous and semantically meaningful.

Table 3: Reconstruction and critical point gradient analysis. Average reconstruction error and average gradient norm with respect to the latent vector $z$ for both the embedding and hypernetwork methods.

| Dataset | Reconstruction error | | Latent vector gradient norm | |
|---|---|---|---|---|
| | Step 1 | Step 2 | Step 1 | Step 2 |
| MNIST | $2.27e^{-2}$ | $3.08e^{-2}$ | $7.14e^{-3}$ | $9.59e^{-4}$ |
| FashionMNIST | $4.55e^{-2}$ | $5.78e^{-2}$ | $2.04e^{-3}$ | $9.67e^{-4}$ |
| ModelNet40 | $7.46e^{-4}$ | $1.88e^{-3}$ | $2.30e^{-4}$ | $5.18e^{-5}$ |
| ShapeNet10 | $7.27e^{-4}$ | $1.21e^{-3}$ | $1.15e^{-4}$ | $2.00e^{-5}$ |
| ScanNet10 | $9.77e^{-4}$ | $4.86e^{-3}$ | $2.66e^{-4}$ | $8.69e^{-5}$ |

## 4.5 EMPIRICAL VALIDATION OF IFT CONDITIONS

Finally, we provide empirical evidence that the conditions required by the IFT hold in our setting. We examine both reconstruction errors and gradient norms. According to equation 8 and equation 10, the implicit function is defined at the critical points of the loss with respect to the latent embeddings $z$; hence, we verify that the average gradient norm of $z$ is very close to zero. Moreover, as indicated in equation 9, the Hessian matrix is full rank only when all data samples are represented exactly. As shown in Table 3, both the training and testing sets achieve sufficiently low reconstruction errors, which can be regarded as negligible. This strongly supports the applicability of the IFT in our framework. While the errors on MNIST and FashionMNIST are slightly higher, they remain lower than those reported in prior work. For example, Inr2Array reports MSE reconstruction losses of $0.0270 \pm 0.0030$ on MNIST and $0.0700 \pm 0.0060$ on FashionMNIST, whereas HyperINR achieves $0.0227$ and $0.0455$, respectively, demonstrating competitive reconstruction quality.

We further check the full rank condition of the Hessian matrix empirically by computing its numerical rank using Singular Value Decomposition (SVD) and the condition number. The results verify that the Hessian is indeed full rank for almost all samples across datasets, as detailed in the A.4.

## 4.6 ABLATION STUDIES

We adopt a unified baseline architecture (Table 11 in A.5) to strictly isolate variables. Across five independent random seeds, HyperINR exhibits negligible variance, with classification accuracy standard deviations of $\pm 0.17\%$ (MNIST) and $\pm 0.69\%$ (FashionMNIST). This confirms that our IFT-based reliably converges to stable solutions regardless of initialization.

A key hypothesis of our work is that INR weights reside on a low-dimensional manifold. Our ablation on latent dimension $z$ (Table 4) empirically validates this. Performance does not improve monotonically with dimension; instead, it peaks at an intrinsic dimension of 20, suggesting that higher dimensionality does not help finding this low-dimensional manifold. In these experiments, we fix the baseline configuration and vary only the target hyperparameter (e.g., $z$) to decouple its

contribution from other architectural factors. A similar trend is observed in the MainNet width (Table 4). The hypernetwork maps $z$ to a specific functional manifold in the weight space. We observe that a moderate width of 64 suffices to parameterize this manifold for MNIST. Increasing the width further (e.g., to 256) yields diminishing results. We provide a detailed breakdown of these ablations and stability diagnostics in A.5.

Table 4: Ablation study on MNIST classification accuracy across varying latent dimensions ($z$) and MainNet widths.

| | Latent Dimension $z$ | | | | | MainNet Width | | | |
|---|---|---|---|---|---|---|---|---|---|
| | 10 | **20** | 64 | 128 | 256 | 32 | **64** | 128 | 256 |
| Accuracy(%) | 81.01 | **98.13** | 97.44 | 96.24 | 91.40 | 97.24 | **98.13** | 97.42 | 94.94 |

## 5 CONCLUSION

With the growing trend of treating weight as a new data modality, we present the HyperINR model as an instrument to ensure that semantic information from data can be reflected in weight representations. Our framework employs a hypernetwork that maps a learnable low-dimensional latent space into the weight space of an Implicit Neural Representation (INR). The learned latent embeddings exhibit natural clustering patterns, and classification experiments indicate that this representation effectively retains the semantics of the original data. Finally, we use the Implicit Function Theorem (IFT) to outline the mapping between data and weight latent representations, offering a potential possibility for or future research in weight space learning.

## LLM USAGE DISCLOSURE

In accordance with the ICLR policy on large language model (LLM) usage, we disclose that an LLM (OpenAIs ChatGPT) was used solely for minor language polishing. This included limited grammar correction and rephrasing for clarity. All research ideas, technical content, analyses, and conclusions were generated entirely by the authors, who remain fully responsible for the papers content. For full transparency, this very disclosure note was also drafted with the help of ChatGPT.

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

Table 5: Dataset statistics

| Dataset | Classes | Train | Test |
|---|---|---|---|
| MNIST | 10 | 60,000 | 10,000 |
| FashionMNIST | 10 | 60,000 | 10,000 |
| ModelNet40 | 40 | 9,843 | 2,468 |
| ShapeNet10 | 10 | 25,512 | 5,103 |
| ScanNet10 | 10 | 6,110 | 1,769 |

## A  APPENDIX

### A.1  DATASET DETAILS

We report dataset statistics including the number of samples, classes, and train/test splits. For 3D point clouds, we uniformly sample 10K queries and compute unsigned distance function (UDF) values by building a KDTree on the fitted point cloud. The nearest-neighbor distance is taken as the UDF value, using the PyTorch3D implementation (Ravi et al., 2020). No augmentation, noise, normalization, or rescaling is applied.

### A.2  NETWORK ARCHITECTURE AND TRAINING

Table 6 summarizes the architectures and training setups used across datasets. We deliberately adopt small and simple networks, keeping the design consistent between 2D and 3D tasks. The latent dimension $z$ is set to either 20 or 30, and all models are trained for 500 epochs in the first step of phase A. The scaling factor for spatial frequency $\omega_0$ used in main net of SIREN architecture (Sitzmann et al., 2020) is 30 for all the data. The paramenter for network stands for the hidden layer size. E.g., "HNet [256,256]" means there are two hidden layers with the size of 256.

Table 6: Architectures and training settings

| Parameter | MNIST | FashionMNIST | ModelNet40 | ShapeNet10 | ScanNet10 |
|---|---|---|---|---|---|
| HNet | [256, 256] | [128, 256] | [256] | [256] | [256] |
| HNet-Heads | - | [256] | [256] | [256] | [256] |
| MNet | [64, 64, 64] | [64, 64, 64] | [64, 128, 64] | [128, 256, 128] | [64, 64, 64] |
| dim($z$) | 20 | 20 | 20 | 20 | 30 |
| HNet LR | $1e^{-4}$ | $1e^{-4}$ | $1e^{-4}$ | $1e^{-4}$ | $5e^{-4}$ |
| $z$ LR | $1e^{-3}$ | $1e^{-3}$ | $1e^{-2}$ | $1e^{-2}$ | $1e^{-2}$ |
| batch size | 2048 | 2048 | 256 | 256 | 256 |
| Classifier | [128, 128, 128] | [128, 128, 128] | [512, 512, 512] | [128, 128, 128] | [256, 512, 256] |

### A.3  ADDITIONAL EXPERIMENTAL DETAILS ON THE COMPARISON WITH INR2ARRAY

We compare our method against the baseline Inr2Array (Zhou et al., 2023b), which originally reports an accuracy of 98.5% on MNIST. Upon reproducing their method using the official code, we obtain a comparable 98.28% accuracy. However, achieving this performance requires a Transformer-based classifier with approximately 22.3M parameters and a training time of ≈26 hours on a single NVIDIA A5000 (24GB RAM). HyperINR utilizes a significantly more compact architecture (≈1M parameters), with the majority of weights concentrated in the hypernetwork heads required to project low-dimensional latent vectors to the INR weight space. Consequently, our joint learning framework converges in under 30 minutes on the same hardwarerepresenting a speedup of over 50×. We also evaluated Inr2Array without auxiliary strategies such as data augmentation, learning rate warmup, and weight decay. Under this controlled setting, Inr2Array's performance drops to 74.30%. We did not scale down the Inr2Array model size, as Transformers typically benefit

from over-parameterization; thus, this comparison favors the baseline. Furthermore, HyperINR achieves lower reconstruction error (MSE 0.0227) compared to Inr2Array (0.0270) without requiring these computational overheads. Notably, when auxiliary training tricks such as data augmentation, warmup, and weight decay are removed from Inr2Array, its MNIST reconstruction error increases to 0.148, highlighting the robustness and efficiency of our method under minimal training enhancements.

## A.4 EMPIRICAL VERIFICATION OF IFT CONDITIONS: HESSIAN RANK

To verify the full rank condition required by our IFT analysis, we numerically computed the Hessians for the full training set (60k) and test set (10k) of both MNIST and FashionMNIST. We analyzed the spectral properties of these Hessians to determine their smallest singular values ($\sigma_{\min}$) and condition numbers ($\kappa$).

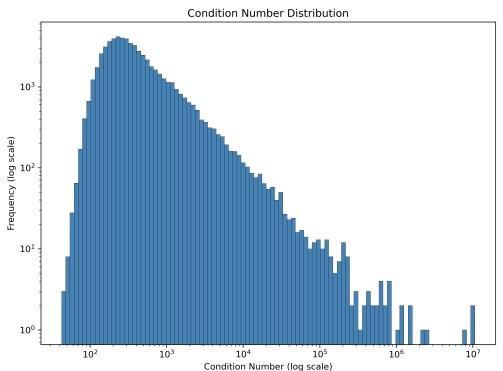 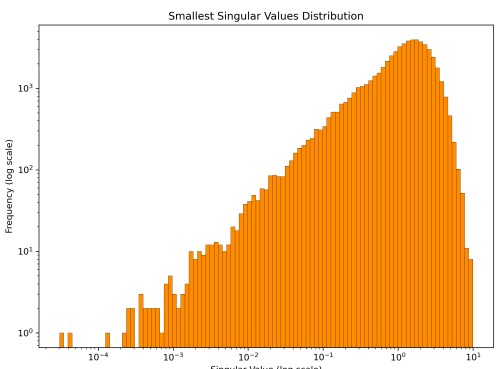

Figure 6: **Hessian analysis on MNIST training data (log-log scale).** The distribution of condition numbers (**left**) shows that the probability mass is concentrated in the well-conditioned range ($10^2 - 10^3$). High condition numbers ($\kappa > 10^5$) appear only as rare, isolated outliers in the long tail. Correspondingly, the smallest singular values (**right**) remain strictly above the float32 machine epsilon ($\sim 1.2 \cdot 10^{-7}$), empirically confirming that the Hessians are full-rank and invertible.

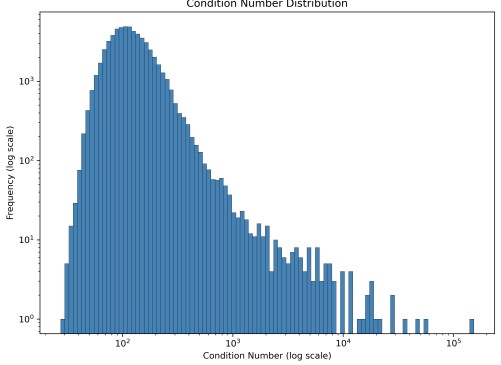 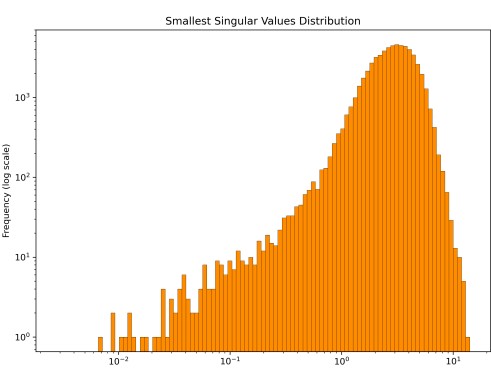

Figure 7: **Hessian analysis on FashionMNIST training data (log-log scale).** The distribution of condition numbers (**left**) shows that the probability mass is concentrated in the well-conditioned range ($10^2 - 10^3$), with negligible outliers. Correspondingly, the smallest singular values (**right**) remain strictly above the float32 machine epsilon ($\sim 1.2 \cdot 10^{-7}$), empirically confirming that the Hessians are full-rank and invertible.

**Numerical non-singularity** A critical requirement for the IFT is that the Jacobian (and by extension the Hessian in our implicit function definition) is non-singular. As shown in Table 8, the smallest singular values for MNIST are approximately $3.05 \times 10^{-5}$ (training set) and $2.53 \times 10^{-5}$ (test set). For FashionMNIST (Table 10), they are $2.79 \times 10^{-3}$ (training set) and $3.95 \times 10^{-5}$ (test set).

Crucially, these values are orders of magnitude above the float32 machine epsilon ($\epsilon \approx 1.2 \times 10^{-7}$). This confirms that the Hessians are numerically full rank across the dataset.

**Condition number** We further analyze the stability of the mapping $g$ via the condition number distribution, detailed in Tables 7 and 9. As illustrated in Fig. 6 and 7, the distribution of condition numbers exhibits a clear concentration of mass in the well-conditioned regime (peaking between $10^2$ and $10^3$). While the theoretical maximum condition numbers reach $\sim 10^7$, the log-scale frequency axis highlights that these are extreme outliers; the curve decays rapidly, rendering the ill-conditioned tail statistically negligible. For instance, in FashionMNIST, less than $0.05\%$ of training samples exceed a condition number of $10^4$. These empirical distributions align closely with our theoretical expectations, suggesting that the local mapping described by the IFT is largely consistent with observations in the weight space for our datasets, with only rare, localized instabilities.

Table 7: Cumulative distribution of Hessian condition numbers on MNIST

| Threshold ($\tau$) | % Samples $> \tau$ (Train) | % Samples $> \tau$ (Test) |
|---|---|---|
| $> 10^1$ | 100.000 | 100.000 |
| $> 10^2$ | 98.002 | 98.710 |
| $> 10^3$ | 16.420 | 20.180 |
| $> 10^4$ | 1.552 | 2.170 |
| $> 10^5$ | 0.168 | 0.290 |
| $> 10^6$ | 0.015 | 0.050 |
| $> 10^7$ | 0.003 | 0.010 |
| $> 10^8$ | 0.000 | 0.000 |
| **Max. $\kappa$** | $1.065 \cdot 10^7$ | $1.227 \cdot 10^7$ |

Table 8: Cumulative distribution of Hessian smallest singular values on MNIST

| Threshold ($\tau$) | % Samples $< \tau$ (Train) | % Samples $< \tau$ (Test) |
|---|---|---|
| $< 10^2$ | 100.000 | 100.000 |
| $< 10^1$ | 100.000 | 100.000 |
| $< 10^0$ | 40.888 | 47.230 |
| $< 10^{-1}$ | 4.617 | 6.280 |
| $< 10^{-2}$ | 0.435 | 0.740 |
| $< 10^{-3}$ | 0.050 | 0.120 |
| $< 10^{-4}$ | 0.005 | 0.010 |
| $< 10^{-5}$ | 0.000 | 0.000 |
| $< 10^{-6}$ | 0.000 | 0.000 |
| **Min. $\sigma_{\min}$** | $3.053 \cdot 10^{-5}$ | $2.534 \cdot 10^{-5}$ |

## A.5 DETAILED ABLATION STUDIES

**Experimental setup and baselines** To strictly control for variables, we employed a unified fixed baseline architecture (detailed in Table 11) across all datasets and independent initialization trials. This standardization ensures that observed deviations in reconstruction fidelity and classification accuracy are attributable solely to intrinsic data complexity or stochastic initialization variance, rather than network architectures or sizes.

**Initialization stability** Across the full spectrum of experimental configurations, we observed that the stochastic variance arising from random initialization is negligible relative to the magnitude of the primary performance metrics. As shown in see Table 12, quantitative analysis of five independent

Table 9: Cumulative distribution of Hessian condition numbers on FashionMNIST

| Threshold ($\tau$) | % Samples $> \tau$ (Train) | % Samples $> \tau$ (Test) |
|---|---|---|
| $> 10^1$ | 100.000 | 100.000 |
| $> 10^2$ | 62.672 | 66.670 |
| $> 10^3$ | 0.427 | 0.850 |
| $> 10^4$ | 0.035 | 0.060 |
| $> 10^5$ | 0.002 | 0.030 |
| $> 10^6$ | 0.000 | 0.010 |
| $> 10^7$ | 0.000 | 0.000 |
| $> 10^8$ | 0.000 | 0.000 |
| **Max. $\kappa$** | $1.532 \cdot 10^5$ | $9.257 \cdot 10^6$ |

Table 10: Cumulative distribution of Hessian smallest singular values on FashionMNIST

| Threshold ($\tau$) | % Samples $< \tau$ (Train) | % Samples $< \tau$ (Test) |
|---|---|---|
| $< 10^2$ | 100.000 | 100.000 |
| $< 10^1$ | 99.950 | 99.980 |
| $< 10^0$ | 3.102 | 5.900 |
| $< 10^{-1}$ | 0.148 | 0.270 |
| $< 10^{-2}$ | 0.007 | 0.030 |
| $< 10^{-3}$ | 0.000 | 0.020 |
| $< 10^{-4}$ | 0.000 | 0.010 |
| $< 10^{-5}$ | 0.000 | 0.000 |
| $< 10^{-6}$ | 0.000 | 0.000 |
| **Min. $\sigma_{\min}$** | $2.789 \cdot 10^{-3}$ | $3.950 \cdot 10^{-5}$ |

trials reveals consistently low standard deviations (typically on the order of $10^{-3}$ to $10^{-4}$), confirming that performance fluctuations remain tightly bounded around the mean. This minimal variance certifies the reproducibility of our results and indicates that the proposed method possesses high robustness to initialization conditions. Consequently, the observed performance deltas in our ablation studies can be confidently attributed to specific architectural interventions rather than stochastic noise, ensuring the statistical validity of our comparative analysis

**Hyperparameter sensitivity** To rigorously evaluate the sensitivity of HyperINR, we implemented a comprehensive ablation protocol (detailed in Table 11) that mirrors the hyperparameter ranges established in prior baselines. Our empirical results demonstrate that HyperINR exhibits high stability across varying architectural configurations. As shown in Table 14 and 15, the model maintains high classification accuracy and low reconstruction error even when the latent dimension is significantly compressed (e.g., $dim(z) = 20$) or when hypernetwork capacity is reduced. This indicates that the structure encoded in weight space is intrinsic to the data structure rather than an artifact of over-parameterization. Table 15 suggests that the optimal INR capacity depends on the complexity of the underlying data. For MNIST, a compact INR architecture (width 64) is sufficient to achieve low reconstruction error ($2.40 \times 10^{-2}$) and high classification accuracy. For FashionMNIST which is complex than MNIST, we observe clear benefits from increased model capacity. Scaling the main INR width to 128 results in improved reconstruction quality and higher downstream classification accuracy compared to smaller baselines. These findings indicate that while HyperINR is robust and matches the network capacity to the data complexity.

Table 11: Baseline configuration for ablation and initialization experiments

| Hyperparameter | Value |
|---|---|
| Latent Dim $z$ | 20 |
| MainNet Width | 64 |
| MainNet Depth | 3 |
| HyperNet Size | [256, 256] |
| HyperNet Heads | 0 |
| Number of epochs | 500 |
| Batch size | 1024 |
| Classifier size | [128, 128, 128] |
| Classifier batch size | 128 |
| Classifier epochs | 150 |

Table 12: Evaluations of Phase A across five random seeds

| Dataset | Recon. Error (Train) | Recon. Error (Test) | Class. Accuracy |
|---|---|---|---|
| MNIST | $(2.42 \pm 0.08) \cdot 10^{-2}$ | $(3.32 \pm 0.11) \cdot 10^{-2}$ | $98.02 \pm 0.17$ |
| FashionMNIST | $(5.12 \pm 0.43) \cdot 10^{-2}$ | $(6.15 \pm 0.40) \cdot 10^{-2}$ | $87.34 \pm 0.69$ |

Table 13: Evaluations of Phase B classifiers across five random seeds

| Dataset | Classification Accuracy |
|---|---|
| MNIST | $98.12 \pm 0.06$ |
| FashionMNIST | $87.53 \pm 0.28$ |

Table 14: Ablation Study on MNIST.

| Hyperparameter | Value | Recon. Train | Recon. Test | Accuracy |
|---|---|---|---|---|
| **Latent Dim** $z$ | 10 | $5.88 \cdot 10^{-2}$ | $1.51 \cdot 10^{-1}$ | 81.01 |
| | 20 | $2.40 \cdot 10^{-2}$ | $3.24 \cdot 10^{-2}$ | 98.13 |
| | 64 | $8.63 \cdot 10^{-3}$ | $1.14 \cdot 10^{-2}$ | 97.44 |
| | 128 | $5.76 \cdot 10^{-3}$ | $6.86 \cdot 10^{-3}$ | 96.24 |
| | 256 | $5.42 \cdot 10^{-3}$ | $7.75 \cdot 10^{-3}$ | 91.40 |
| | 512 | $4.45 \cdot 10^{-3}$ | $6.35 \cdot 10^{-3}$ | 83.82 |
| | 1024 | $3.98 \cdot 10^{-3}$ | $7.17 \cdot 10^{-3}$ | 87.39 |
| **MainNet Width** | 32 | $2.81 \cdot 10^{-1}$ | $1.05 \cdot 10^{-1}$ | 97.24 |
| | 64 | $2.40 \cdot 10^{-2}$ | $3.24 \cdot 10^{-2}$ | 98.13 |
| | 128 | $1.88 \cdot 10^{-2}$ | $3.77 \cdot 10^{-2}$ | 97.42 |
| | 256 | $1.27 \cdot 10^{-2}$ | $5.49 \cdot 10^{-2}$ | 94.94 |
| **MainNet Depth** | 1 | $1.38 \cdot 10^{-1}$ | $1.49 \cdot 10^{-1}$ | 96.31 |
| | 2 | $3.24 \cdot 10^{-2}$ | $3.79 \cdot 10^{-2}$ | 98.22 |
| | 3 | $2.40 \cdot 10^{-2}$ | $3.24 \cdot 10^{-2}$ | 98.13 |
| | 4 | $3.61 \cdot 10^{-2}$ | $4.86 \cdot 10^{-2}$ | 96.50 |
| | 5 | $2.76 \cdot 10^{-2}$ | $5.35 \cdot 10^{-2}$ | 94.70 |
| | 6 | $2.17 \cdot 10^{-2}$ | $5.01 \cdot 10^{-2}$ | 97.36 |
| **HyperNet Size** | [128, 128] | $2.65 \cdot 10^{-2}$ | $3.97 \cdot 10^{-2}$ | 97.18 |
| | [256, 256] | $2.40 \cdot 10^{-2}$ | $3.24 \cdot 10^{-2}$ | 98.13 |
| | [512, 512] | $2.10 \cdot 10^{-2}$ | $3.36 \cdot 10^{-2}$ | 97.46 |
| **HyperNet Heads** | 0 | $2.40 \cdot 10^{-2}$ | $3.24 \cdot 10^{-2}$ | 98.13 |
| | 128 | $4.64 \cdot 10^{-2}$ | $5.29 \cdot 10^{-2}$ | 97.41 |
| | 256 | $2.63 \cdot 10^{-2}$ | $3.96 \cdot 10^{-2}$ | 94.41 |
| | 512 | $2.19 \cdot 10^{-2}$ | $3.73 \cdot 10^{-2}$ | 95.65 |
| | 1024 | $1.89 \cdot 10^{-2}$ | $3.22 \cdot 10^{-2}$ | 96.68 |

Table 15: Ablation Study on FashionMNIST.

| Hyperparameter | Value | Recon. Train | Recon. Test | Accuracy |
|---|---|---|---|---|
| **Latent Dim** $z$ | 10 | $6.92 \cdot 10^{-2}$ | $1.09 \cdot 10^{-1}$ | 73.15 |
| | 20 | $5.72 \cdot 10^{-2}$ | $6.58 \cdot 10^{-2}$ | 86.23 |
| | 64 | $2.56 \cdot 10^{-2}$ | $3.17 \cdot 10^{-2}$ | 87.18 |
| | 128 | $1.91 \cdot 10^{-2}$ | $2.57 \cdot 10^{-2}$ | 83.52 |
| | 256 | $1.85 \cdot 10^{-2}$ | $2.70 \cdot 10^{-2}$ | 79.84 |
| | 512 | $1.52 \cdot 10^{-2}$ | $2.65 \cdot 10^{-2}$ | 64.01 |
| | 1024 | $1.41 \cdot 10^{-2}$ | $2.38 \cdot 10^{-2}$ | 72.82 |
| **MainNet Width** | 32 | $8.33 \cdot 10^{-2}$ | $9.50 \cdot 10^{-2}$ | 84.64 |
| | 64 | $5.72 \cdot 10^{-2}$ | $6.58 \cdot 10^{-2}$ | 86.23 |
| | 128 | $3.80 \cdot 10^{-2}$ | $5.50 \cdot 10^{-2}$ | 88.61 |
| | 256 | $2.29 \cdot 10^{-2}$ | $7.30 \cdot 10^{-2}$ | 83.64 |
| **MainNet Depth** | 1 | $1.13 \cdot 10^{-1}$ | $1.38 \cdot 10^{-1}$ | 75.29 |
| | 2 | $5.80 \cdot 10^{-2}$ | $7.38 \cdot 10^{-2}$ | 82.88 |
| | 3 | $5.72 \cdot 10^{-2}$ | $6.58 \cdot 10^{-2}$ | 86.23 |
| | 4 | $5.43 \cdot 10^{-2}$ | $6.79 \cdot 10^{-2}$ | 84.31 |
| | 5 | $4.60 \cdot 10^{-2}$ | $6.11 \cdot 10^{-2}$ | 88.11 |
| | 6 | $4.14 \cdot 10^{-2}$ | $6.29 \cdot 10^{-2}$ | 86.68 |
| **HyperNet Size** | [128, 128] | $5.13 \cdot 10^{-2}$ | $6.06 \cdot 10^{-2}$ | 86.60 |
| | [256, 256] | $5.72 \cdot 10^{-2}$ | $6.58 \cdot 10^{-2}$ | 86.23 |
| | [512, 512] | $3.97 \cdot 10^{-2}$ | $6.05 \cdot 10^{-2}$ | 83.65 |
| **HyperNet Heads** | 0 | $5.72 \cdot 10^{-2}$ | $6.58 \cdot 10^{-2}$ | 86.23 |
| | 128 | $7.14 \cdot 10^{-2}$ | $1.17 \cdot 10^{-1}$ | 68.72 |
| | 256 | $5.00 \cdot 10^{-2}$ | $5.99 \cdot 10^{-2}$ | 88.07 |
| | 512 | $4.47 \cdot 10^{-2}$ | $8.18 \cdot 10^{-2}$ | 80.58 |
| | 1024 | $3.74 \cdot 10^{-2}$ | $7.88 \cdot 10^{-2}$ | 75.86 |

