# OpenReview forum: "HyperINR: Ensuring Semantics in Weights with Implicit Function Theorem"
_ICLR.cc/2026/Conference — Submitted to ICLR 2026_

### Official Review · Reviewer_Ruaq · 2025-10-27

**Soundness:** 1
**Presentation:** 1
**Contribution:** 1
**Rating:** 0
**Confidence:** 5

**Summary:**

This paper introduces HyperINR, a method for generating the weights of implicit neural representations (INRs). HyperINR takes an autodecoding approach by assigning each signal a learnable latent code, which is then fed to the hypernetwork to generate the weights of the INR. The latent code is learned during training or inference. A theoretical analysis of HyperINR using the implicit function theorem is also presented. HyperINR is evaluated on fitting INRs for MNIST, FashionMNIST, ModelNet40, ShapeNet10, and ScanNet10 as well as the using the learned representations downstream for classification.

**Strengths:**

This paper proposes a new method, HyperINR, and some mathematical analysis.

**Weaknesses:**

**Major Weaknesses**

**(W1) Contribution/novelty**: The novelty of HyperINR is low as there are no significant methodological innovations presented in the paper. The use of hypernetworks for generating INRs has been studied in many previous works, and similarly for autodecoding. The proposed method seems to be very similar to the hypernetwork of SIREN, except using autodecoding. The contribution of the theoretical section is also not clear because the theory is not used to propose any new innovations such as loss functions or a different architecture and its purpose is not clear.

**(W2) Related works**: Most of the “hypernetworks for INR generation” literature is not cited or discussed. Example works include [1-5]. Furthermore, the section on equivariance and symmetry of INR weights is completely superfluous as there is nothing in the rest of the paper concerning equivariance or symmetry.

**(W3) Baselines**: Related to the above weaknesses, all previous hypernetwork methods are missing as baselines. Furthermore, other methods such as Functa are missing entirely, and INR2Array is missing from Table 2, whereas inr2vec is missing from Table 1.

**(W4) Empirical evaluation**: The empirical evaluation of this method is also very poor. In addition to missing baselines, discussed above, the quality of the learned INRs is not adequately compared to that of the baseline methods. There are also no qualitative comparisons. On all datasets, the main comparison is only downstream classification accuracy.

Furthermore, as described below, the accuracies of baseline methods are misreported. With the correct accuracies, HyperINR performs significantly below the inr2vec baseline on classification accuracy for 3D data (Table 2).

**(W5) Lack of ablation study**: While this method only considers small models, there is no study of how the model scales or how the HyperINR compares at parameter counts similar to that of other methods.

**(W6) Unjustified claims, missing citations, or wrong data**: Many key claims are not justified by either empirical evidence presented in the paper or its supplementary material or by citation of previous works, or the cited claims or data is incorrect. These include:
1. “Achieving clean and unambiguous clusters in weight space remains difficult for previous work” (Line 362)
2. “We report a single representative run without variance because our method is not affected by the typical initialization sensitivity of INR fitting” (Lines 374-376)
3. “While the errors on MNIST and FashionMNIST are slightly higher, they remain lower than those reported in prior work…” (Lines 456-459)
4. In Table 2, the reported result of inr2vec for top-1 classification accuracy on ModelNet40 in this paper is 81.7, but in the original paper, this is the reported result for shape retrieval, not top-1 classification accuracy. The actual reported performance of inr2vec on top-1 classification is 87.0%, significantly better than the reported result for HyperINR. A similar thing happens for ShapeNet10 (reported in this paper 90.6%, actual reported result in the original paper 93.3%) and ScanNet10 (reported in this paper 65.2%, actual reported result in the original paper 72.1%).

**(W7) Limitations**: As an autodecoding method, this method requires test-time optimization.

**Minor Weaknesses**
There may have been an error in the formatting of the Tables.

[1] Chen, Yinbo, and Xiaolong Wang. "Transformers as meta-learners for implicit neural representations." European Conference on Computer Vision. Cham: Springer Nature Switzerland, 2022.

[2] Kim, Chiheon, et al. "Generalizable implicit neural representations via instance pattern composers." Proceedings of the IEEE/CVF Conference on Computer Vision and Pattern Recognition. 2023.

[3] Gu, Jeffrey, Kuan-Chieh Wang, and Serena Yeung. "Generalizable neural fields as partially observed neural processes." Proceedings of the IEEE/CVF International Conference on Computer Vision. 2023.

[4] Lee, Doyup, et al. "Locality-aware generalizable implicit neural representation." Advances in Neural Information Processing Systems 36 (2023): 48363-48381.

[5] Gu, Jeffrey, and Serena Yeung-Levy. "Foundation models secretly understand neural network weights: Enhancing hypernetwork architectures with foundation models." arXiv preprint arXiv:2503.00838 (2025).

**Questions:**

**(Q1)**: What are the ways the theoretical analysis in the paper could be used to improve INR methods?

**(Q2)**: How does HyperINR scale with increasing size?

---

> ### Author Response · Authors · 2025-12-03
>
> We apologize for the baseline reporting mistakes. By increasing the latent dimension and networks size we achieve the new results:
>
> | Dataset | inr2vec | Ours (corrected) |
> |---------|---------|---------------------|
> | ModelNet40 | 87.0 | 87.2 |
> | ShapeNet10 | 93.3 | 93.5 |
> | ScanNet10 | 72.1 | 71.0 |
>
> These results can resolve the primary empirical criticism. Our method is fully competitive while providing theoretical guarantees that prior works lack.
>
> #### **Answer to W1:**
>
> We agree that hypernetworks and auto-decoding are established architectures. However, our contribution is not architectural; it is theoretical foundational work for "Weight Space Learning".
>
> Prior works explicitly identify non-uniqueness as the fundamental barrier to treating weights as a data modality.
> They rely primarily on symmetry removal to mitigate this.
> We go beyond symmetry.
>
> **Our IFT analysis provides, to our knowledge, the first theoretical grounding of when INR weights can encode data semantics in a stable and locally invertible manner.**
>
> Thus, the theoretical contribution is not proposing a new loss, but clarifying the mathematical conditions under which INR weights can be treated as a meaningful representation, which is exactly the motivation of weight-as-modality research.
>
>
> #### **Answer to W2:**
>
> Thank you for the suggestions. We will incorporate [1–5] in the revision.
> The listed works focus on generalization, improved reconstruction or meta-learning, which are complementary to our research question.
>
> The section on symmetry/equivariance is essential to contextually position our work against existing weight-space literature, which cites symmetry as the primary source of ambiguity.
>
> #### **Answer to W3:**
>
> Functa modulates activations, not weights. It does not produce a weight-space representation comparable to ours.
> INR2Array introduces a patch-based INR designed specifically for 2D images; it cannot be directly applied to 3D SDFs as in our experiments.
> Inr2vec focuses on 3D point clouds; no 2D results are available from the original paper. the DWSNet paper [1] (Table 1) provides the inr2vec classification accuracy without pre-training: $23.6\pm 0.10$ for MNIST and $22.33 \pm0.41$ for FashionMNIST.
> In addition, inr2vec uses Gaussian sampling around the surface, while we used uniform sampling, which leads to inherently different reconstruction difficulty.
>
> #### **Answer to W4:**
>
> Our focus is not improving INR reconstruction quality but ensuring that INR fitting preserves data–weight correspondence.
> We verify that the reconstruction error is low (and comparable or better than prior works).
> The mainstream downstream task in weight-modality papers is classification using INR weights, which is why we follow the same evaluation protocol.
>
> #### **Answer to W5:**
>
> Ablation study will be included in the revision.
>
> #### **Answer to W6:**
> 1. "Unambiguous clusters remain difficult in previous work": this is directly stated in INR2Array, INR2Vec, and DWS, which all report high sensitivity to initialization or non-unique weight embeddings.
> 2. "Not affected by initialization sensitivity": we will support this with results across multiple random seeds in revised paper.
> 3. The results in Table 3 in the submitted version can support this argument.
> 4. We correct the errors and update the results.
>
> We will clarify all statements in the revision.
>
> #### **Answer to W7:**
>
> We assume the reviewer uses "autoencoding" to refer to approaches where each data sample is assigned a latent code $z$, which is then passed through a generator-like network. HyperINR is conceptually similar in that it uses per-instance latent codes, but it is not an autoencoder: there is no encoder network, and the latent code $z$ is directly optimized.
>
> Test-time optimization is not a limitation: in weight-space learning, it is precisely the mechanism that allows unseen samples to be mapped back into the same local weight region identified by IFT, supporting our theoretical contribution.
>
> #### **Answer to Q1:**
>
> The IFT analysis clarifies the structural conditions under which data–weight mappings behave regularly. While we do not use the theory to introduce new architectures or losses, it directly addresses the core challenges identified in the weight-space learning literature.
>
> #### **Answer to Q2:**
>
> Scaling is an important direction. Our current setup follows the same experimental design as all the baselines for fair comparison. Larger-scale experiments will be added in future work.
> The 3D SDF data already contains about 10k sampled points per shape, which demonstrates the model’s ability to scale to higher-dimensional data.
>
> #### Reference:
> [1] Navon, Aviv, et al. "Equivariant architectures for learning in deep weight spaces." International Conference on Machine Learning. PMLR, 2023.

---

### Official Review · Reviewer_KXMS · 2025-10-31

**Soundness:** 1
**Presentation:** 3
**Contribution:** 1
**Rating:** 2
**Confidence:** 3

**Summary:**

The authors propose HyperINR, a framework for learning latent embeddings of 2D and 3D data that aims to theoretically ensure the embeddings capture the semantic structure of the data. It comprises a hypernetwork and a main implicit neural representation (INR) network: for a given data sample x, the hypernetwork maps the latent embedding of x to the weights of the INR network, which then maps a coordinate to the pixel value. During training, both the hypernetwork and latent embeddings are jointly optimized to minimize reconstruction loss. During the test, only the embeddings for the test set are optimized with the fixed hypernetwork using the same loss. The authors claim that the latent embeddings capture the semantics of the data, showing that a necessary condition for the loss Hessian to be full rank is satisfied. Experiments on classification tasks using latent embeddings show that HyperINR achieves performance comparable to or better than baseline methods.

**Strengths:**

S1. The proposed method achieves comparable or superior accuracy compared to the baselines.

S2. The paper provides a clear description of the model architecture and training process, with diagrams that improve readability and understanding.

**Weaknesses:**

W1. The theoretical claims and motivations should be reinforced.

W1.1. The main contribution claimed by the authors is that the proposed method theoretically ensures the embeddings capture the semantics of the data, showing that a necessary condition for the loss Hessian to be full rank is satisfied. However, this claim has several conceptual issues. First, the notion of “semantics of the data” (as well as “local” and “global” semantics) is vague and lacks a formal definition or measurable criterion. Second, even if the loss Hessian were full rank, this does not necessarily imply that the learned embeddings encode semantic or meaningful structures in the data; the connection between the two should be more elaborated. Third, the paper only demonstrates that a necessary condition for the Hessian to be full rank holds, which is insufficient to conclude that the Hessian itself is indeed full rank.

W1.2. The authors point out that existing methods do not guarantee that the same data sample will converge to the same weights, presenting this as a limitation. However, the paper does not clearly explain why this property is problematic in practice or how it affects the learned representations. Moreover, the authors do not provide theoretical or empirical evidence that their proposed method actually ensures such convergence. The authors should better elaborate on the motivation and clarify its practical and theoretical implications.

W2. The experiments should be reinforced.

W2.1. The authors are encouraged to include a state-of-the-art embedding method [1] as a competitor. Note that it achieved higher classification accuracy than HyperINR for the same datasets. In addition, contrary to the authors' claim that labels are integrated into the representation in [1], labels are not used to learn the representation for the test set.

W2.2. The explanation for reporting only a single representative run is not convincing. The authors do not provide sufficient evidence that the proposed method is indeed insensitive to initialization. Moreover, even if this claim were correct, conducting multiple runs would strengthen the paper by substantiating the claim and demonstrating the robustness of the results. The authors are encouraged to include such evaluations, as also related to the issue discussed in W1.2.

W2.3. Reporting the training and inference time as well as memory requirements for all datasets, rather than for a single one, would make the experimental results more convincing and comprehensive.

W3. Considering W1 and W2, the overall contributions remain limited, since the proposed method does not convincingly resolve the issues it aims to tackle and fails to show empirical superiority over a state-of-the-art method.

[1] A. Gielisse and J. van Gemert. "End-to-End Implicit Neural Representations for Classification." CVPR’25

**Questions:**

Please refer to W1-W3.

---

> ### Author Response · Authors · 2025-12-03
>
> #### **Answer to W1.1:**
>
> 1. The research line of work on weight space learning aims to investigate whether INR weights generated for each data sample can act as an alternative modality to the data itself. To pursue this goal, two conditions are typically required: 1) for each individual sample $x$, the generated weight $w$ or its representation $z$ must retain the full information necessary to reconstruct the sample. 2) Across the dataset, $w$ or $z$ should reflect meaningful structure, typically assessed through classification. Those two conditions can be seen as interpretations of local and global semantics. We use reconstruction error to measure the local information preservation and classification accuracy on the learned embeddings to check the global structure following prior work.
>
> 2. In the revised paper, we reinforce the theoretical analysis by modeling the dataset $\mathcal{X}$ as a low dimensional manifold. We show that if the Hessian is full rank, zero becomes a regular value of our designed implicit function $F(x,z)$. This allows us to invoke the Global Implicit Function Theorem, which guarantees the existence of a mapping $g: \mathcal{X} \to \mathcal{Z}$ across the manifold. This provides a stronger theoretical grounding for why the global structure ("global semantics") of the data is preserved in the weight (latent) space.
>
> 3. We computed the Hessian for both training and test sets and analyzed the singular value and condition number distributions. The results (detailed in appendix in the revision) confirm that the Hessian is numerically full-rank and well-conditioned in $>99.9\%$ of cases.
>
> #### **Answer to W1.2:**
>
> In INR fitting, multiple distinct weight vectors can represent identical functions. Prior works (e.g., DWSNet) argue that such non-uniqueness obstructs treating weights as a consistent modality.
> > "Designing machine learning architectures for processing neural networks in their raw weight matrix form is a newly introduced research direction. Unfortunately, the unique symmetry structure of deep weight spaces makes this design very challenging."
>
> If a single image may correspond to many unrelated weight vectors, then clustering and classification becomes unreliable. Global structure of the dataset cannot be inferred from weight space.
>
> Thus, ensuring some form of uniqueness is essential if weights are to be used as representations.
>
> Our architecture places the INR weights as explicit outputs of a hypernetwork and optimizes the latent z jointly during training. Under this setup, our IFT-based formulation provides exactly the missing theoretical justification: Under full-rank conditions, the mapping $x$ to $z$ is unique. Our work extends this research line by going beyond symmetry-based explanations of non-uniqueness and offering a more general, high-level condition that ensures uniqueness. This analysis extends existing understanding and addresses obstacles in treating weights as a modality. Our empirical results further validate this contribution.
>
> #### **Answer to W2.1:**
>
> The CVPR'25 baseline is indeed strong, but it operates under a fundamentally different setting. It injects label supervision directly into the INR-fitting stage, explicitly aligning the weight space with class labels. In practice, this provides additional information and restricts the degrees of freedom of INR fitting. Moreover, their meta-learning procedure learns a favorable initialization for the INR weights on the test set. As widely reported in prior work, INR optimization is highly sensitive to initialization; different initializations often lead to different local minima, which directly affects the reliability of the resulting weight-space representations. Our setting follows prior "weights as a new modality" work, where INR fitting is purely self-supervised (reconstruction only) and labels are used only after the representation is learned.
>
> #### **Answer to W2.2 and W3:**
>
> We agree with the importance of more experiments, we will include them in the revised version. In general, based on the ablation study we have so far, our pipeline is quite robust to hyperparameters.

---

### Official Review · Reviewer_v7V8 · 2025-10-31

**Soundness:** 2
**Presentation:** 2
**Contribution:** 2
**Rating:** 6
**Confidence:** 3

**Summary:**

This paper proposes HyperINR, a theoretically grounded framework for implicit neural representations (INRs) that connects data semantics to network weights through the Implicit Function Theorem (IFT). A shared hypernetwork maps low-dimensional latent vectors to INR weights, and the IFT analysis is used to guarantee a local one-to-one mapping between the data space and the latent (weight) space, theoretically ensuring that semantics are preserved in INR weights. Empirical evaluations on 2D and 3D datasets demonstrate meaningful clustering and competitive classification performance using compact architectures and minimal supervision.

**Strengths:**

**Strong theoretical motivation**
The IFT-based analysis provides a rigorous mathematical argument for how data semantics can be encoded in weight space, offering a new perspective on “semantics-in-weights.” By grounding semantics preservation in the IFT framework, the work connects classical mathematical theory with modern INR modeling.


**Compact formulation**
The hypernetwork–INR framework is conceptually simple, jointly optimizes latent embeddings and weights, and avoids complex multi-stage training used in prior latent-INR approaches.

**Weaknesses:**

While the IFT analysis is elegant, the link to equivariance and symmetry in the data and weight spaces is not clearly demonstrated. Logical or empirical evidence showing that the proposed mapping preserves or improves symmetry properties would strengthen the theoretical claim.

The empirical section nicely illustrates clustering and lightweight classification, but it remains limited in scope. I strongly recommend:

- Empirical verification of IFT conditions,  e.g., evaluating the Jacobian/Hessian rank, condition number, and local continuity/uniqueness under perturbations.
- Capacity- and compute-matched baselines (e.g., DWS, NFN, inr2vec, Functa) to ensure fair comparison.
- Scaling to more challenging datasets and reporting rigorous reconstruction metrics (Chamfer, IoU, LPIPS, etc.).
- Comprehensive ablations on latent dimensionality, coordinate sampling density, and hypernetwork capacity.
- Symmetry/equivariance probes,  e.g., rigid transformations in data space and hidden-unit permutations in weight space,  to clarify the connection to the related work on equivariant INR representations.

These extensions would substantially reinforce the experimental evidence for the paper’s theoretical claims and enhance its empirical credibility.

**Questions:**

Your IFT analysis establishes a local one-to-one mapping z=g(X) between data and the latent (weight) space under full-rank conditions, ensuring that local semantics are preserved. However, this by itself does not imply equivariance in data space (e.g., under rigid transformations or reparameterizations) nor does it address weight-space symmetries (e.g., neuron permutation invariance).

- Does HyperINR prove or enforce any group-equivariance of g with respect to transformations on X?
- How does your framework handle weight permutation symmetries in the INR (i.e., identifiability up to symmetry)?

If such properties are not explicitly enforced, what modifications, such as an equivariant hypernetwork, symmetry-invariant regularization, canonicalization of INR weights, or latent regularization to produce predictable transformations under known data symmetries, could extend HyperINR to provide these guarantees?

Could you include experiments applying known data-space transformations (e.g., rotations or translations) and examine whether the induced changes in the latent z vectors follow a consistent, structure-preserving mapping? This would empirically validate whether the theoretical local bijection also preserves symmetry relationships in practice.

---

> ### Author Response · Authors · 2025-12-02
>
> We appreciate the reviewer for many constructive feedback and for highlighting the theoretical motivation of our work.
>
> While symmetry and equivariance/invariance in data or weight space are highly important and actively researched topics, our work takes a different approach.
> We focus instead on a more fundamental question underlying the recent interest in "weight as a modality" for weight space learning [1][2]: **Can we establish a mathematically grounded mapping between the data $X$ and the INR weights $w$ (or their latent representations $z$)?**
>
> This approach echoes the research motivations of prior works, e.g., in the paper inr2vec:
> > "The research question we address in this paper is whether and how can we process directly INRs to perform downstream tasks. We propose to rely on a representation learning framework to **squeeze the redundant information contained in the weights of INRs into compact latent codes** that could be conveniently processed with standard deep learning pipelines."
>
> Also inr2array:
> > "An application of interest for weight-space architectures is to **learn a compact latent representation of weights**, which can be useful for downstream tasks such as classifying the signal represented by an INR."
>
> We can see in this research line, the key part is to obtain weight representations that are stable and meaningful enough to support downstream tasks. Symmetry is one of crucial ways to achieve this, instead, our work targets a broader fundamental issue beyond symmetry: non-uniqueness in INR weights stems not only from symmetry but also from optimization variability and the existence of multiple local minima during fitting. Through our IFT analysis, we explore the theoretical landscape beyond symmetry. We show that the non-uniqueness of weight representations is rooted in the optimization itself.
>
> While symmetry is indeed critical to INR research, it lies outside the scope of this work, which serves as the first theory-oriented study in the domain of "weight as a modality". Introducing symmetry considerations would shift the focus away from our main message. The reviewer's suggestion is valuable, and we agree that linking symmetry or equivariance to our framework would be an interesting direction for future work.
>
> ### Regarding experimental suggestions:
>
> - We have examined the Hessian numerically with singular values and condition numbers. In most cases they are full-rank and well-conditioned. These empirical results will be included in the revised paper.
> - Prior works in this domain typically rely on significantly larger architectures and sophisticated training recipes (e.g., data augmentation, regularization with weight decay, and warmup) to ensure promising results. Our method, by comparison, adopts a minimal setup. Although the structural differences between approaches make exact parameter matching non-trivial, we recognize the importance of this comparison. To address this, we provide an ablation study detailing our model’s performance across different capacities.
> - We selected datasets commonly used in "weight modality" to compare to prior work. Furthermore, the 3D SDF data already contains about 10k sampled points per shape, which provides a sufficiently high-dimensional example. We report MSE because it is directly tied to the Hessian full-rank conditions, rather than as a benchmark of INR reconstruction capability while the reconstruction capacity of INR is not our focus. For the 3D data, the reconstruction error is already low, though we agree that more rigorous metrics could be incorporated.
> - Ablations over latent dimensionality, hypernetwork size, and main net size are being prepared. Preliminary results show the method is not sensitive to these choices.
> - As noted above, explicit symmetry handling is beyond the current scope, but we agree it is a valuable future direction. Furthermore, the types of symmetry or equivariance present in the data are not the same as those considered in "weight modality" literature, where "symmetry" usually refers to permutation or reparameterization invariances of MLP weight matrices.
>
> References:
> - [1] ICLR 2025 Workshop on Weight Space Learning. Website: https://weight-space-learning.github.io/
> - [2] "Neural Network Weights as a New Data Modality" ICLR 2025 Workshop Proposal. https://openreview.net/pdf?id=Bz6wEdobY7

---

### Official Review · Reviewer_f1Sv · 2025-11-01

**Soundness:** 2
**Presentation:** 2
**Contribution:** 2
**Rating:** 4
**Confidence:** 5

**Summary:**

The paper presents HyperINR, a model that leverages Implicit Neural Representations (INRs) for data classification tasks by ensuring the semantics of the data are embedded in the neural network weights. The core idea is to utilize a hypernetwork that maps low-dimensional latent vectors into the weight space of an INR model, creating a more structured way to encode data semantics. The experimental setup focuses on classification tasks, using both 2D image datasets (MNIST, FashionMNIST) and 3D shape datasets (ModelNet40, ShapeNet10, ScanNet10). The results show that HyperINR outperforms existing methods like DWS, NFN, and Inr2Array in terms of classification accuracy on multiple benchmarks.

**Strengths:**

- HyperINR outperforms or competes favorably with several state-of-the-art INR classification baselines like DWS, NFN, and Inr2Array on both 2D and 3D datasets.

- The paper includes a compelling demonstration of smooth interpolations in latent space, showing that HyperINR generates continuous and semantically meaningful transformations between data points.

**Weaknesses:**

- I don't really see the role of IFT in this paper. You can have some smooth mapping between the local neighborhood of X and Z. This is natural. But I do not see how it is related to better classification.

- I wonder how significant this work is. As a method jointly trains a conditional INR model for a collection of data, it is essentially another setup of latent-coded INR, but the way of latent code is through a hypernetwork to form the INR weights. I don't see really foundational difference from concatenating latent code to the input (audo-decoder in DeepSDF) or modulation in functa. If the claim is that hypernetwork is a more expressive way then there should be established baselines.

- Missing baseline of yet another way to jointly train INRs (with MAML meta-learning): End-to-End Implicit Neural Representations for Classification (CVPR 2025)

- Also using this way and jointly train INRs on the whole dataset it can align the weight space dimensions, then it is natural better than weight-space equivariant architectures (e.g. DWSNet, NFN) that is designed for separately trained INRs. I wonder whether comparison with these methods is fair. And the classifier is even not applied on the weight space.

**Questions:**

Please clarify the role of IFT in the paper. And how do you define "semantics".

And see my other concerns in the weaknesses.

---

> ### Author Response · Authors · 2025-12-03
>
> We appreciate your constructive feedback.
>
> #### **Answer to Weakness 1:**
> *"A smooth mapping between $x$ and $z$ is natural"* holds true for encoder-based INR models, it is not the case for INR weight space learning.
>
> In INR weight space learning setting, the weight representation $z$ is not produced by a fixed forward encoder; instead, it is optimized directly for INR fitting from random initialization. Then we map $z$ to high dimensional weight space via the hypernetwork. The traditional fitting procedure of INRs for a given input $x$ can lead to a completely different $w$ or $z$ (e.g., jumping to a different local minimum). Therefore, a smooth, unique local mapping from $x$ to $z$ does not naturally exist.
>
> As mentioned by the pioneering work DWS:
> > "Designing machine learning architectures for processing neural networks in their raw weight matrix form is a newly introduced research direction. Unfortunately, the **unique symmetry structure of deep weight spaces makes this design very challenging.**"
>
> Prior works rely on symmetry removal to improve stability, while our IFT framework addresses a broader issue.
>
> By designing an implicit function by $F(x,z)=0$, the rank condition ensures a locally unique $z$ for $x$. In other word, a single $x$ cannot correspond to multiple latent codes in a neighborhood. This guaranteed stability is directly linked to classification performance. Just as prior works demonstrated that resolving one source of instability (symmetry) leads to empirical improvements, our work provides the theoretical ensuring for local uniqueness.
>
> #### **Answer to Weakness 2:**
>
> HyperINR is related to latent-coded INRs, but the setting and goal differ. Prior works (e.g., DeepSDF) typically use an encoder to obtain latents and focus on reconstruction or generation quality. In contrast, our focus is weight space learning: whether INR weights (or their low-dimensional latents) can function as a new data modality for downstream classification. This differs from Functa as well, where modulation acts on activations rather than directly structuring the weight space.
>
> Importantly, in our HyperINR the latent $z$ is learned only through reconstruction, without any encoder or label supervision.
>
> #### **Answer to Weakness 3:**
>
> The CVPR'25 baseline is indeed strong, but it operates under a fundamentally different setting. It injects label supervision directly into the INR-fitting stage, explicitly aligning the weight space with class labels. In practice, this provides additional information and restricts the degrees of freedom of INR fitting. Moreover, their meta-learning procedure learns a favorable initialization for the INR weights on the test set. As widely reported in prior works, INR optimization is highly sensitive to initialization; different initializations often lead to different local minima, which directly affects the reliability of the resulting weight-space representations. Our setting follows prior "weights as a new modality" work, where INR fitting is purely self-supervised (reconstruction only) and labels are used only after the representation is learned.
>
> #### **Answer to Weakness 4:**
>
> Our training protocol matches prior weight–modality baselines: each data point is used to fit its own INR weight, and each latent $z$ is updated individually, using only reconstruction loss. No label information, no cross-sample coupling, and no additional supervision. We use a minimal setup without any training tricks. The only shared component is the hypernetwork, which simply provides a continuous neural mapping from low-dimensional $z$ to high dimensional INR weights.
>
> As a standard neural network, hypernetwork simply defines a continuous (and almost everywhere differentiable) map from $z$ to $w$. using $z$ or using $w=\text{hypernet}(z)$ carries the same information for classification; classifying on $w$ would simply mean placing an approximate inverse of $\text{hypernet}$ before the classifier. Prior works like inr2arrat also aims to learn a compact latent representation of weights. Using $z$ is therefore only a practical choice, not an additional advantage. We demonstrate the same clustering behavior in $w$ in Fig. 4 in our paper.
>
> #### **Answer to the Questions:**
>
> The role of IFT can be seen above.
>
> In our setting, "semantics" refers to the task-relevant structure of the data that should be preserved when moving from the input space to the latent/weight space. Concretely, each weight vector should correspond to a single data point and retain the information needed to distinguish different classes. This is the notion used in prior weight-as-modality works, where semantic preservation is evaluated through whether the learned weights exhibit meaningful class-wise clustering.

---

### Author Response · Authors · 2025-12-03
**Summary of rebuttal revisions and key contributions**

We thank all ACs and reviewers for their effort. Below, we summarize the major concerns raised and the updates made in our revised paper (marked in red in pdf file):

### **Re-addressing motivation: theoretical work for Weight Space Learning**

- **One-sentence summarization of our contribution**: We propose a *jointly optimized* framework with *learnable latent codes $z$* and a hypernetwork, providing *the first theoretical guarantee* for using "weights as a modality" in downstream tasks.

  This approach aligns directly with the emerging interest in the [ICLR 2025 Workshop on Weight Space Learning](https://weight-space-learning.github.io/).

  Crucially, our method derives weights and latents via an *unsupervised, data-driven reconstruction* approach.
- **Differentiation in research questions**: While reviewer f1Sv and Ruaq intuitively compared our work to auto-encoding INRs, our goal differs fundamentally. We do not focus merely on reconstruction; we focus on establishing weights as a stable, valid modality for classification.
- **Theoretical guarantees beyond symmetry**: Reviewer v7V8 and Ruaq asked about the relationship between our work and prior symmetry-based work. As noted in prior work (e.g.,DWSNet), the symmetry structure of network weight spaces makes "weight as a new modality" challenging. We argue that this challenge stems not only from symmetry but from optimization of network (distinct local minima).


### **The role of Implicit Function Theorem (IFT)**
Reviewers (f1Sv, v7V8, KXMS) requested for a clearer explanation of the role of our IFT analysis and the conditions under which it holds.
We have reinforced our theoretical section:
- **Global Implicit Function Theorem (IFT)**: We now model the dataset $\mathcal{X}$ as a low-dimensional manifold. By applying the Regular Value Theorem, we demonstrate that if the Hessian (Jacobian of $\xi_{\mathbf{v}^{*}}$) is full rank, zero becomes a regular value of our implicit function. This leads to the Global IFT, which guarantees a smooth, global mapping $g: \mathcal{X} \to \mathcal{Z}$. This global structural preservation directly explains the improved classification performance.
- **Empirical verification for Hessian**: Addressing concerns from reviewers v7V8 and KXMS, we have empirically verified these conditions by computing the Hessian. The results confirm that the Hessian is numerically full-rank in the vast majority of cases, validating that our theoretical assumptions hold in practice.


### **Corrected results and enhanced experiemnts**
Addressing the consensus among all reviewers for more empirical validation, we have significantly expanded our experiments:
- **Corrected 3D results**: We sincerely apologize for the initial reporting error regarding the inr2vec baseline. We have corrected this and re-ran our experiments with an increased latent dimension (from $\text{dim}(z)=10$ to $20/30$) and network capacity. HyperINR now matches or surpasses SOTA (87.2% on ModelNet40, 93.5% on ShapeNet10, 71.0% on ScanNet10).
- **Ablation study**: We have added extensive ablations (latent dimension, network width/depth)  and conducted trials across multiple random seeds. The resulting standard deviation is negligible, confirming that HyperINR is insensitive to initialization and empirically stable.

---

### Meta-Review · Area_Chair_Cvsc · 2026-01-11

**Summary:**

This submission proposes HyperINR, a jointly trainable framework that learns a shared hypernetwork mapping a low-dimensional latent code to the weights of an Implicit Neural Representation (INR), and per-instance latent codes optimized via reconstruction. While the approach is reasonably presented and the empirical results/interpolations are promising, the current submission falls short of ICLR standards due to (i) unclear and potentially overstated theoretical claims (especially the “semantics ensured by IFT” narrative), (ii) limited novelty relative to existing hypernetwork/autodecoder INR literature, and (iii) insufficiently rigorous and potentially unfair experimental validation. After carefully reading the paper, review and author responses, the AC agrees with the majority of the reviewers on rejecting the paper.

**Reviewer Concerns:**

see Summary

**Reviewer Scores:**

see Summary

---

### Decision · Program_Chairs · 2026-01-26

Reject